# Distinct regulation of dopamine D2S and D2L autoreceptor signaling by calcium

Stephanie C Gantz[1], Brooks G Robinson[1], David C Buck[2,3], James R Bunzow[1], Rachael L Neve[4], John T Williams[1], Kim A Neve[2,3]*

[1]Vollum Institute, Oregon Health & Science University, Portland, United States; [2]Research Service, VA Portland Health Care System, United States Department of Veterans Affairs, Portland, United States; [3]Department of Behavioral Neuroscience, Oregon Health & Science University, Portland, United States; [4]Department of Brain and Cognitive Sciences, Massachusetts Institute of Technology, Cambridge, United States

**Abstract** D2 autoreceptors regulate dopamine release throughout the brain. Two isoforms of the D2 receptor, D2S and D2L, are expressed in midbrain dopamine neurons. Differential roles of these isoforms as autoreceptors are poorly understood. By virally expressing the isoforms in dopamine neurons of D2 receptor knockout mice, this study assessed the calcium-dependence and drug-induced plasticity of D2S and D2L receptor-dependent G protein-coupled inwardly rectifying potassium (GIRK) currents. The results reveal that D2S, but not D2L receptors, exhibited calcium-dependent desensitization similar to that exhibited by endogenous autoreceptors. Two pathways of calcium signaling that regulated D2 autoreceptor-dependent GIRK signaling were identified, which distinctly affected desensitization and the magnitude of D2S and D2L receptor-dependent GIRK currents. Previous in vivo cocaine exposure removed calcium-dependent D2 autoreceptor desensitization in wild type, but not D2S-only mice. Thus, expression of D2S as the exclusive autoreceptor was insufficient for cocaine-induced plasticity, implying a functional role for the co-expression of D2S and D2L autoreceptors.

*For correspondence: nevek@ohsu.edu

**Competing interests:** The authors declare that no competing interests exist.

**Reviewing editor**: Sacha B Nelson, Brandeis University, United States

## Introduction

Central dopamine transmission coordinates reinforcement learning, including recognition of reward-predictive stimuli and initiation of goal-directed movements. Natural rewards, reward-predictive cues, and drugs of abuse elicit a rapid increase in dopamine release from dopamine axon terminals and somatodendritic sites within the ventral midbrain. Dopamine release is negatively regulated by the activation of dopamine D2 autoreceptors on somatodendritic and axon terminals (reviewed in *Ford, 2014*). Loss of D2 autoreceptor-mediated inhibition results in elevated extracellular dopamine and is associated with perseverative drug-seeking, enhanced motivation for food, and novelty-induced hyperactivity (*Marinelli and White, 2000*; *Marinelli et al., 2003*; *Bello et al., 2011*; *Anzalone et al., 2012*; *Holroyd et al., 2015*). Chronic D2 autoreceptor activation impairs the formation of dopamine- and glutamate-releasing axon terminals (*Fasano et al., 2010*). Thus, D2 autoreceptors regulate structural and functional plasticity of dopamine neurons and are essential in limiting impulsivity and reward-seeking behaviors.

A prominent feature of somatodendritic D2 autoreceptors is their activation of G protein-coupled inwardly rectifying potassium (GIRK) channels, resulting in inhibition of action potential firing and subsequent dopamine release. During prolonged activation, desensitization of D2 autoreceptors reduces the D2 autoreceptor-dependent GIRK current. A component of desensitization is dependent on intracellular calcium (*Beckstead and Williams, 2007*). Single or repeated exposure to drugs of

**eLife digest** Dopamine is an important component of the brain's reward system and is commonly referred to as a 'feel-good' chemical. It is mainly released from neurons in the brain in response to natural rewards, such as food or sex, and following exposure to, or in anticipation of, certain drugs of abuse (including cocaine).

Dopamine-releasing neurons also sense dopamine, and just like someone can change the volume of their voice by hearing themselves speak, dopamine neurons regulate how much dopamine is released based on how much dopamine they sense. This feedback system is known as autoinhibition. These neurons sense dopamine when it binds to, and activates, so-called 'dopamine D2 receptors' on their cell surface. But not all D2 receptors are alike. Instead there are two variants called D2S and D2L.

Previous studies have shown that D2 receptor signaling in dopamine neurons is altered by the concentration of calcium ions inside these cells. Furthermore, exposure to cocaine and other drugs is known to change how these calcium ions regulate D2 receptor signaling. Now, Gantz et al. have used mice that produce only a single variant of the D2 receptor (either D2S or D2L) in their dopamine neurons to uncover similarities and differences between the two variants. The experiments show that localized increases in calcium ion concentration make D2S less capable of autoinhibition, like D2 receptors in neurons from wild type mice, without affecting autoinhibition by D2L.

In further experiments, some of these mice were given cocaine before D2 receptor signaling was assessed. In dopamine neurons from wild type mice, a single exposure to cocaine eliminates the calcium-dependent regulation; thus, cocaine treatment causes a D2L-like response. In contrast, cocaine treatment did not affect the calcium-dependent regulation when only one variant of the D2 receptor was present. This implies that dopamine neurons must have both D2S and D2L receptors before the drug can induce changes in D2 receptor signaling. These findings also challenge the long-held view that the D2S receptor is the predominant form involved in autoinhibition. The next challenge is to determine how cocaine induces an apparent switch from D2S to D2L and the implications of this switch for the development of cocaine addiction.

abuse modifies D2 autoreceptor function (*Henry et al., 1989*; *Wolf et al., 1993*; *Jones et al., 2000*; *Marinelli et al., 2003*; *Gantz et al., 2013*; *Madhavan et al., 2013*), including a loss of the calcium-dependent component of D2 autoreceptor-GIRK desensitization (*Perra et al., 2011*). The mechanism(s) that underlie acute desensitization and drug-induced plasticity of D2 autoreceptor-mediated inhibition remain incompletely characterized.

There are two splice variants of the D2 receptor, which differ by a 29-amino acid insert in the third intracellular loop in D2-Long (D2L) that is absent in D2-Short (D2S). Biased expression of D2S or D2L receptors alters behavioral responses to drugs of abuse (*Smith et al., 2002*; *Bulwa et al., 2011*) and has been associated with drug addiction in human studies (*Sasabe et al., 2007*; *Moyer et al., 2011*). Functionally distinct roles for D2S and D2L receptors have been proposed based on characterization of mice lacking D2L (*Usiello et al., 2000*; *Wang et al., 2000*) or D2S (*Radl et al., 2013*). Behavioral and biochemical studies have designated D2L as the postsynaptic receptor expressed on non-dopaminergic medium spiny neurons in the basal forebrain and D2S as the autoreceptor (*Khan et al., 1998*; *Usiello et al., 2000*; *Lindgren et al., 2003*). However, both D2S and D2L receptors are expressed in dopamine neurons and function as somatodendritic autoreceptors (*Khan et al., 1998*; *Jomphe et al., 2006*; *Jang et al., 2011*; *Neve et al., 2013*; *Dragicevic et al., 2014*). Biochemical studies indicate that D2S receptors internalize and desensitize more readily than D2L receptors (*Liu et al., 1992*; *Itokawa et al., 1996*; *Ito et al., 1999*; *Morris et al., 2007*; *Thibault et al., 2011*), but acute desensitization and drug-induced plasticity of D2S and D2L receptor-dependent GIRK currents have not been characterized. Using virus-mediated expression of the D2 receptor splice variants in D2 receptor knockout mice, this study reveals that D2S but not D2L receptor-dependent GIRK signaling exhibited calcium-dependent desensitization. Manipulations of pathways involved in D2 autoreceptor desensitization had distinct actions on D2S and D2L receptor-dependent GIRK currents. Lastly, a single in vivo cocaine exposure removed the calcium-dependent component of D2 autoreceptor-GIRK desensitization in wild type mice, but not D2S-only mice; thus, the expression of D2S as the exclusive autoreceptor was insufficient

for drug-induced plasticity. Taken together, the results of this study imply a physiological role for the co-expression of D2S and D2L autoreceptors.

## Results

### Both D2S and D2L can function as autoreceptors

To examine the ability of D2S and D2L receptors to activate a GIRK conductance, single isoforms were expressed in midbrain dopamine neurons. $Drd2^{-/-}$ mice received bilateral injections of an adeno-associated viral (AAV) vector generating either rat D2S or D2L receptor and GFP expression, as previously described (*Neve et al., 2013*). Infected neurons in brain slices containing the substantia nigra *pars compacta* (SNc) were identified by GFP visualization. Whole-cell patch clamp recordings were made from SNc dopamine neurons using an internal solution containing the calcium chelator, BAPTA (10 mM), as used previously (*Neve et al., 2013*). Application of a saturating concentration of the D2 receptor agonist, quinpirole (30 µM), produced an outward current that was reversed by application of the D2 receptor antagonist, sulpiride (600 nM, *Figure 1A,B*). There was no difference in the peak amplitude of quinpirole-induced currents mediated by D2S and D2L receptors (*Figure 1B*). In the continued presence of agonist, D2 autoreceptors desensitize resulting in a decline in the agonist-induced outward current (*Beckstead and Williams, 2007*; *Perra et al., 2011*). The decline in the quinpirole-induced current mediated by D2S and D2L receptors was indistinguishable (*Figure 1A,C*).

In the SNc, dopamine release from neighboring dopamine neurons elicits D2 receptor-mediated inhibitory postsynaptic currents (IPSCs) through the activation of GIRK channels (*Beckstead et al., 2004*; *Gantz et al., 2013*). D2S and D2L receptors mediate kinetically-identical IPSCs following electrically stimulated dopamine release (*Neve et al., 2013*). Stimulus-independent dopamine release also occurs, resulting in spontaneous D2 receptor-mediated IPSCs (*Gantz et al., 2013*). In slices from mice infected with either D2S or D2L, spontaneous IPSCs were abolished by application of sulpiride (600 nM, *Figure 1D,E*). The durations of D2S, D2L, and wild type D2 receptor-mediated spontaneous IPSCs were identical (*Figure 1F*, [from *Gantz et al., 2013*, WT: 515 ± 17 ms, $n$ = 76 sIPSCs]). Amplitude and frequency of spontaneous IPSCs are affected by the level of D2 receptor expression and dopamine synthesis (*Gantz et al., 2013*, *2015*). Since these parameters may be influenced by variegated viral infection, the amplitude and frequency of D2S- and D2L-sIPSCs were not compared. Taken together, the results confirm previous work indicating that D2S and D2L can serve as autoreceptors at somatodendritic dopamine synapses (*Neve et al., 2013*).

### Calcium entry promotes desensitization of D2 autoreceptors in wild type dopamine neurons

Desensitization in the GIRK current induced by D2 receptor agonists is affected by intracellular calcium buffering. Weak calcium buffering with intracellular EGTA (0.025–0.4 mM) results in greater decline in the GIRK current induced by D2 receptor agonists, without affecting the decline in the GIRK current induced by GABA_B receptor agonists (*Beckstead and Williams, 2007*; *Perra et al., 2011*). These results were confirmed in wild type mice using internal solutions containing either EGTA (0.1 mM, EGTA internal) or BAPTA (10 mM, BAPTA internal). Application of quinpirole (10 µM) or the GABA_B agonist, baclofen (30 µM), resulted in outward currents that declined in the continued presence of agonist (*Figure 2A,E*). The peak amplitudes of the quinpirole- and baclofen-induced currents were larger when using BAPTA internal than EGTA internal (*Figure 2A,B,E,F*). The quinpirole-induced current desensitized more quickly with EGTA internal compared with experiments using BAPTA internal (*Figure 2A,C,D*). This calcium-dependent desensitization was specific to the D2 receptor since the decline in baclofen-induced current was not dependent on the internal solution (*Figure 2E,G,H*). Thus, as reported previously (*Beckstead and Williams, 2007*; *Perra et al., 2011*), D2 autoreceptors exhibited a calcium-dependent desensitization that resulted in a larger decline in the D2 autoreceptor-dependent current when intracellular calcium was buffered with a low concentration of EGTA.

EGTA and BAPTA have a similar affinity for calcium but differ in the kinetics of binding. This property is frequently used to characterize the distance between a calcium source and a calcium sensor. Buffering with EGTA allows calcium to diffuse farther (microdomain) than BAPTA, which limits calcium spread (nanodomain) from a calcium source. However, the concentrations of EGTA and BAPTA used in this study may also result in different levels of resting free calcium (*Adler et al., 1991*). To determine whether the difference in D2 autoreceptor desensitization observed with the two

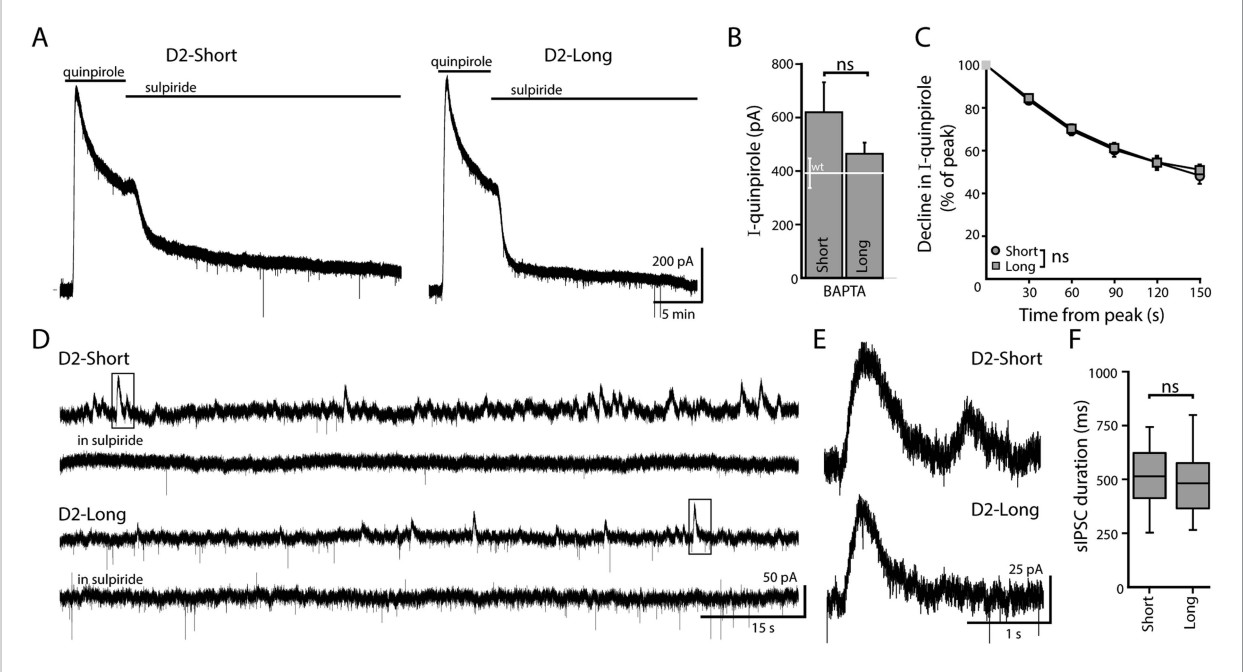

**Figure 1**. When virally expressed in midbrain dopamine neurons, D2S and D2L function as autoreceptors. (**A**) Representative traces of whole-cell voltage clamp recordings, using a BAPTA-containing internal solution, of the outward current in D2S and D2L neurons induced by bath application of quinpirole (30 μM), which was reversed by sulpiride (600 nM). (**B**) The amplitude of quinpirole-induced currents in D2S and D2L neurons using BAPTA internal did not differ ($n$ = 12–14, unpaired $t$ test), shown in reference to the amplitude of the quinpirole-induced currents in WT neurons (white line). (**C**) There was no difference between D2S and D2L in the decline of the D2 receptor-dependent current in the continued presence of quinpirole using BAPTA internal (two-way ANOVA). (**D**, **E**) Representative traces of spontaneous D2-sIPSCs mediated by D2S and D2L receptors, blocked by sulpiride. Inset boxes are shown enlarged in (**E**). The frequency and amplitude of D2S- and D2L-sIPSCs were not analyzed since these parameters may be influenced by the expression of D2 receptors in presynaptic dopamine neurons, which cannot be confirmed. (**F**) The duration of D2S-sIPSCs and D2L-sIPSCs did not differ ($n$ = 84–100 sIPSCs, Mann–Whitney $U$ test). ns indicates not significant.

internals was explained by resting free calcium concentration, the level of free calcium in the BAPTA internal was increased to 300 nM by addition of $CaCl_2$ (7.37 mM) (BAPTA+$Ca^{2+}$, see 'Materials and methods'). The peak amplitude and the decline in the quinpirole-induced current recorded with BAPTA+$Ca^{2+}$ internal were not different from measurements recorded with BAPTA alone (*Figure 3A,B*). Interestingly, BAPTA+$Ca^{2+}$ internal decreased the peak amplitude of the baclofen-induced current significantly relative to the amplitude recorded with BAPTA internal (*Figure 3C*), and was not different from the current measured with EGTA internal (*Figure 3C*). The decline in the baclofen-induced current was unaffected by BAPTA+$Ca^{2+}$ (*Figure 3D*).

To verify that the resting free calcium was increased using BAPTA+$Ca^{2+}$ internal, the positive modulator of the small conductance calcium-activated potassium channel (SK), NS309 (10 μM) was applied. Although NS309 did not produce a current using either BAPTA or EGTA internal, it caused an outward current with BAPTA+$Ca^{2+}$ internal (*Figure 3—figure supplement 1A,B*). The NS309-induced current was reversed by the SK channel blocker apamin (200 nM). Thus, the BAPTA+$Ca^{2+}$ internal increased resting free calcium.

Taken together, the results indicate that the resting free calcium had differential actions on D2 and $GABA_B$ receptor-dependent GIRK currents. The magnitude of the $GABA_B$ receptor-dependent current was sensitive to resting free calcium, but the decline in current was independent of resting free calcium. The decline in D2 autoreceptor-dependent current was dominated by the spatial regulation of intracellular calcium, not resting free calcium.

## Desensitization of D2S- but not D2L-GIRK currents is calcium-dependent

Recordings were made from dopamine neurons that expressed D2S or D2L receptors using an internal solution containing EGTA (0.1 mM). Application of quinpirole (30 μM) produced an outward

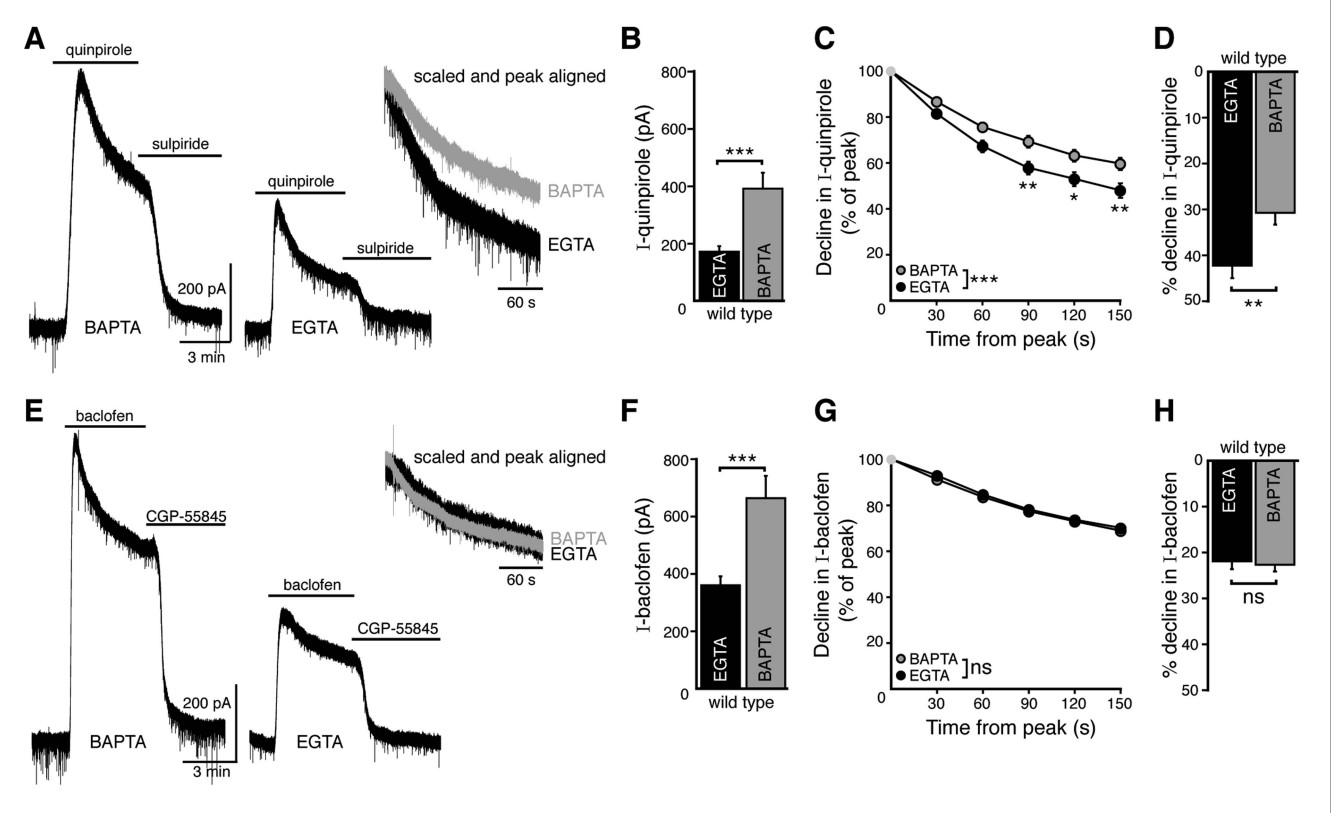

**Figure 2**. Weak intracellular calcium buffering reveals calcium-dependent desensitization of D2 autoreceptor-dependent GIRK currents in wild type dopamine neurons. (**A**) Representative traces of whole-cell voltage clamp recordings of the outward current induced by bath application of quinpirole (10 µM) that was reversed by sulpiride (600 nM), using a BAPTA or EGTA-containing internal solution. (**B**) The amplitude of the quinpirole-induced current was larger using BAPTA than EGTA internal (*n* = 15 each, unpaired *t* test). (**C, D**) The decline in quinpirole-induced current was faster using EGTA internal compared to BAPTA (**C**: two-way ANOVA followed by Bonferroni, **D**: unpaired *t* test) (**E**) Representative traces of whole-cell voltage clamp recordings of the outward current induced by bath application of baclofen (30 µM) which was reversed by CGP-55845 (200 nM), using BAPTA or EGTA internal. (**F**) The amplitude of the baclofen-induced current was larger using BAPTA than EGTA internal (*n* = 14–16, unpaired *t* test). (**G, H**) There was no difference in the decline in baclofen-induced current recorded with EGTA and BAPTA internals (**G**: two-way ANOVA, **D**: unpaired *t* test). ns indicates not significant, *$p < 0.05$, **$p < 0.01$, ***$p < 0.001$.

current that declined and was reversed by sulpiride (600 nM). In D2S neurons, the decline using EGTA internal was faster than the decline using BAPTA internal (*Figure 4A,D*, *Figure 4—figure supplement 1A*). In contrast, in D2L neurons, the decline with EGTA and BAPTA internal was not different (*Figure 4A,D*, *Figure 4—figure supplement 1B*). The insensitivity of the decline in D2L neurons to calcium buffering resulted in significantly more desensitization of D2S than D2L with EGTA internal (*Figure 4B*). The peak amplitude of the quinpirole-induced currents in D2S and D2L neurons was not different (*Figure 4C*), indicating the difference between D2S and D2L is unlikely to be due to differences in the level of expression of D2 receptors. Application of the GABA_B agonist, baclofen (30 µM), produced an outward current that was reversed by the GABA_B antagonist, CGP-55845 (200 nM). The peak amplitude of the baclofen-induced current was not different among D2S (EGTA: 259 ± 28 pA; BAPTA: 531 ± 94 pA), D2L (EGTA: 277 ± 29 pA; BAPTA: 520 ± 43 pA), D2-KO (AAV-GFP-only, EGTA: 279 ± 54 pA; BAPTA: 664 ± 80 pA), and wild type dopamine neurons (p > 0.05). There was also no change in the decline in the baclofen-induced current in D2S- or D2L-expressing neurons (*Figure 4E*).

To minimize potential confounds of ectopic D2 receptor expression in non-dopamine neurons in the midbrain, the calcium-sensitivity of D2S receptor-GIRK desensitization was validated using a transgenic D2S mouse line, generated by a cross between TH-hD2S (*Gantz et al., 2013*) and D2 receptor knockout mice. In this line, the expression of Flag-tagged human D2S receptors depends on the tyrosine hydroxylase promoter (*Figure 4—figure supplement 2*). In slices from these mice, quinpirole (10 µM)

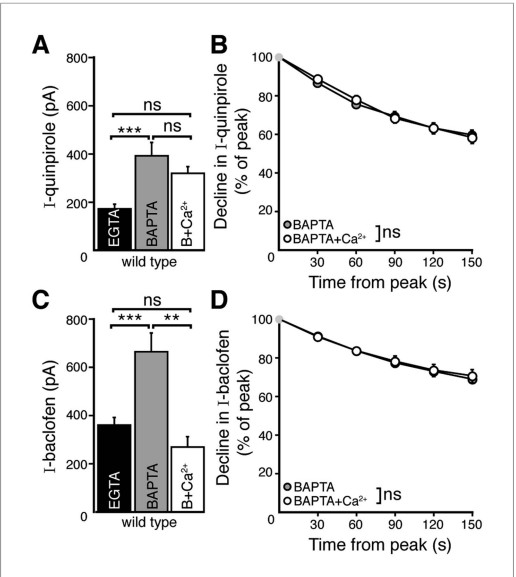

**Figure 3**. Increasing resting free internal calcium does not enhance desensitization of D2 autoreceptor-dependent GIRK currents. (**A**) The amplitude of the quinpirole (10 μM)-induced current using BAPTA+Ca$^{2+}$ internal solution was not different from the amplitudes using BAPTA or EGTA internal solutions ($n = 7$–15, ANOVA followed by Bonferroni). (**B**) Increasing resting free calcium with BAPTA+Ca$^{2+}$ had no effect on the decline in quinpirole-induced current (two-way ANOVA). (**C**) Increasing resting free calcium with BAPTA+Ca$^{2+}$ internal decreased the amplitude of the baclofen (30 μM)-induced current, making it no greater than the amplitude recorded using EGTA internal (ANOVA followed by Bonferroni). (**D**) Increasing resting free calcium with BAPTA+Ca$^{2+}$ had no effect on the decline in baclofen-induced current (two-way ANOVA). Additional experiments that demonstrate BAPTA+Ca$^{2+}$ internal increased resting free calcium can be found in *Figure 3—figure supplement 1*. ns indicates not significant, **$p < 0.01$, ***$p < 0.001$.

The following figure supplement is available for figure 3:

**Figure supplement 1**. The positive modulator of the SK channel, NS309, produces an outward current when using the BAPTA+Ca$^{2+}$ internal solution.

produced larger outward currents using BAPTA internal compared to EGTA internal (*Figure 4F*). The currents were significantly larger than those recorded in wild type dopamine neurons (*Figures 2B, 4F*), indicating overexpression of D2 receptors. Despite the overexpression, the magnitude of the decline in the quinpirole-induced current was similar to wild type (*Figures 2C,D, 4H*). Also consistent with wild type, the decline in the quinpirole-induced current using EGTA internal was significantly faster than the decline using BAPTA internal (*Figure 4G,H*). Taken together, these results indicate that D2S but not D2L receptor-GIRK signaling exhibited calcium-dependent desensitization.

## Calcium signaling regulates D2 autoreceptor-activated GIRK conductance

### Intracellular calcium stores

Endoplasmic calcium stores contribute to calcium-dependent desensitization of D2 autoreceptor-GIRK signaling (*Perra et al., 2011*). Cyclopiazonic acid (CPA) disrupts the sarco/endoplasmic reticulum calcium-ATPase leading to rapid depletion of intracellular calcium stores (*Ford et al., 2010*). Brain slices were exposed to CPA (10 μM) >20 min prior to making recordings. As shown previously in wild type dopamine neurons, CPA reduced the decline in the quinpirole-induced current using EGTA internal and had no effect when using BAPTA internal (*Figure 5A*). CPA did not change the decline in the baclofen-induced current recorded with either internal (*Figure 5B*). In D2S neurons, CPA also reduced the decline in the quinpirole-induced current, but the decline in the quinpirole-induced current in D2L neurons was not changed (*Figure 5C*). These results indicate that calcium release from intracellular stores contributed to D2S but not D2L receptor-GIRK desensitization.

In wild type neurons with EGTA internal, CPA had no significant effect on the magnitude of the maximal current produced by bath application of quinpirole (control: 172 ± 19 pA, $n = 15$; CPA: 194 ± 22 pA, $n = 13$, p = 0.46, unpaired $t$ test). In a previous study, sub-saturating D2 receptor-dependent currents repeatedly produced by pressure ejection of dopamine are augmented by bath application of CPA for 10–20 min (*Perra et al., 2011*). Therefore, the effect of CPA was examined on submaximal dopamine currents produced by iontophoretic application of dopamine once every 50 s (I-DA). In wild type neurons using an EGTA internal, CPA (10 μM) significantly augmented I-DA (*Figure 5D*). While CPA rapidly depletes intracellular calcium stores, the CPA-induced augmentation of I-DA did not plateau until >15 min (*Figure 5—figure supplement 1*). CPA also significantly augmented I-DA in D2S and D2L neurons (*Figure 5E* and *Figure 5—figure supplement 1*). However, the magnitude of the increase was significantly greater for D2L receptor-dependent currents than D2S (*Figure 5E*). Thus, depletion of calcium from intracellular stores differentially increased D2S and D2L receptor-dependent GIRK signaling.

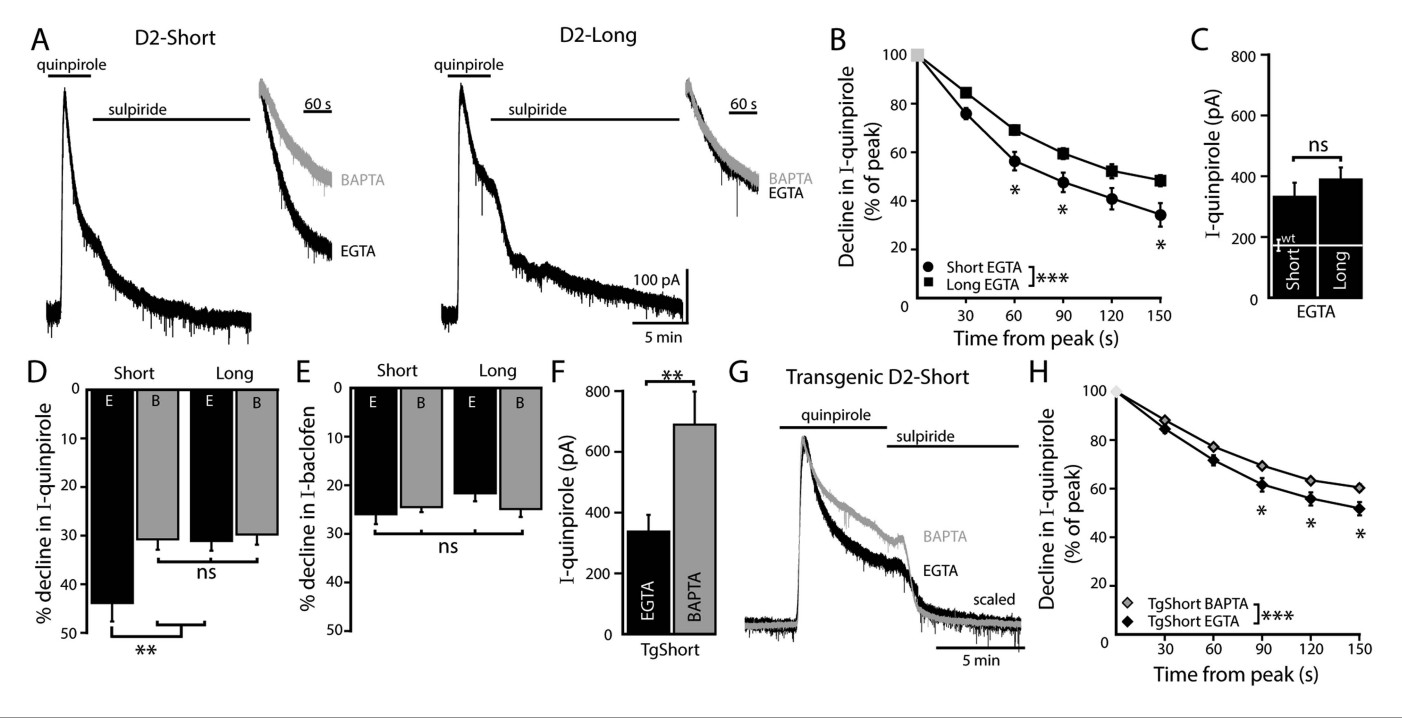

**Figure 4**. D2S but not D2L receptor-GIRK currents exhibit calcium-dependent desensitization. (**A**) Representative traces of whole-cell voltage clamp recordings of the outward current in D2S and D2L neurons induced by bath application of quinpirole (30 μM) which was reversed by sulpiride (600 nM), using EGTA internal, compared with the BAPTA trace shown in *Figure 1A* (scaled and peak-aligned). (**B**, **D**) Using EGTA internal, the decline in quinpirole-induced current was greater in D2S than D2L neurons (**B**: two-way ANOVA followed by Bonferroni, **D**: n = 16 each, one-way ANOVA followed by Fisher's LSD). (**C**) The amplitude of quinpirole-induced currents in D2S and D2L neurons using EGTA internal did not differ (n = 16–17, unpaired t test), shown in reference to the amplitude of the quinpirole-induced currents in WT neurons (white line). (**D**) In D2S neurons the decline in quinpirole-induced current was greater using EGTA internal compared to BAPTA, but not in D2L neurons (n = 12–16, one-way ANOVA followed by Fisher's LSD). The time course of the decline can be found in *Figure 4—figure supplement 1*. (**E**) There was no difference in the decline in baclofen-induced current recorded with EGTA or BAPTA internal in either splice variant (n = 11–19, one-way ANOVA). (**F**) In neurons from transgenic D2S mice, the amplitude of the quinpirole-induced current was larger using BAPTA than EGTA internal (n = 7–8, unpaired t test). (**G**, **H**) Representative scaled and peak-aligned traces of whole-cell voltage clamp recordings from neurons from transgenic D2S mice of the outward currents induced by bath application of quinpirole (10 μM), which were reversed by sulpiride. The decline in quinpirole-induced current was greater using EGTA internal compared to BAPTA (two-way ANOVA followed by Bonferroni). ns indicates not significant, *p < 0.05, **p < 0.01, ***p < 0.001.

The following figure supplements are available for figure 4:

**Figure supplement 1**. Time course of desensitization of D2 receptor splice variant-GIRK currents.

**Figure supplement 2**. Expression and labeling of Flag-D2S receptors in dopamine neurons.

## L-type calcium channels

In SNc dopamine neurons, calcium entry via somatodendritic low-voltage-activated L-type calcium channels occurs during tonic 'pacemaker' action potential firing, creating oscillations of elevated cytosolic calcium in the somatodendritic compartment (*Chan et al., 2007*; *Puopolo et al., 2007*; *Hage and Khaliq, 2015*). L-type calcium channels may be involved in D2 autoreceptor desensitization, (*Goldberg et al., 2005*; *Guzman et al., 2010*; *Dragicevic et al., 2014*), but no studies have directly measured D2 autoreceptor-dependent GIRK signaling. To determine if L-type calcium channels regulate D2 autoreceptor-dependent GIRK signaling, brain slices were exposed to the L-type calcium channel blocker, isradipine (300 nM), >20 min prior to making recordings. In wild type dopamine neurons, isradipine did not significantly change the decline in the quinpirole-induced current using either EGTA or BAPTA internal (*Figure 6A*). Isradipine also had no effect on the decline in baclofen-induced current (*Figure 6B*). However, isradipine reduced the decline in the quinpirole-induced current in D2S

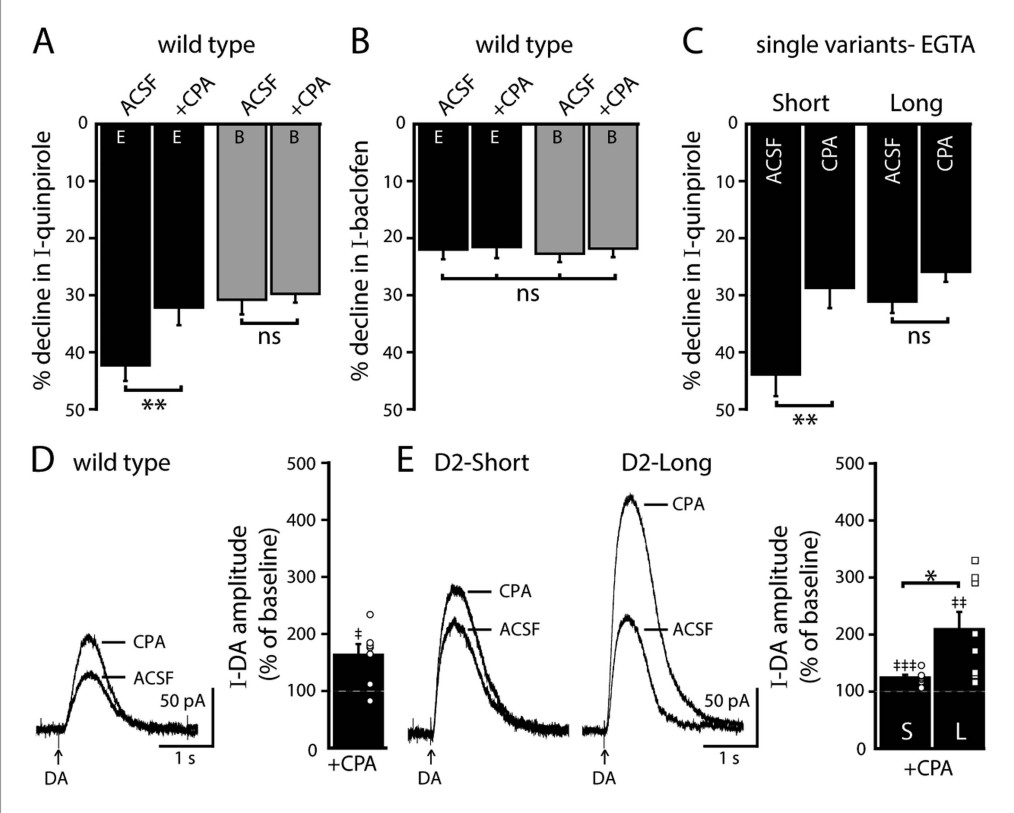

**Figure 5**. Depleting intracellular calcium stores differentially modifies D2S and D2L receptor-dependent GIRK conductance. (**A**) In wild type neurons, CPA (10 µM, > 20 min) reduced the decline in the quinpirole-induced current using EGTA internal and the effect was prevented with the use of BAPTA internal ($n = 11$–$15$, one-way ANOVA followed by Fisher's LSD). (**B**) CPA had no effect on the decline in baclofen-induced current recorded with EGTA or BAPTA internal in wild type neurons ($n = 14$–$16$, one-way ANOVA). (**C**) In D2S, but not D2L neurons, CPA reduced the decline in quinpirole-induced current using EGTA internal ($n = 6$–$16$, one-way ANOVA followed by Fisher's LSD). (**D**, **E**) Submaximal D2 receptor-dependent outward currents were produced by iontophoretic application of dopamine once every 50 s while recording with EGTA internal (I-DA, arrows). (**D**) CPA (10 µM, 25–30 min) augmented I-DA in wild type neurons, shown in representative averaged traces (left) and grouped data (right, $n = 7$). (**E**) CPA augmented I-DA in D2S and D2L neurons, shown in representative averaged traces (left) and grouped data (right). The augmentation by CPA was greater in D2L than D2S neurons ($n = 7$–$8$, unpaired $t$ test). The time course of the CPA-induced augmentation of I-DA can be found in *Figure 5—figure supplement 1*. Baseline: mean amplitude of six I-DAs preceding CPA application, ns indicates not significant, *$p < 0.05$, **$p < 0.01$, ***$p < 0.001$, and ‡ indicates significance over baseline (within-group comparison, paired $t$ tests).

The following figure supplement is available for figure 5:

**Figure supplement 1**. Prolonged CPA application enhances D2 receptor-dependent currents produced by exogenous dopamine.

neurons, without affecting the decline in D2L neurons (*Figure 6C*). Taken together, the results suggest that calcium influx via L-type calcium channels was not involved in desensitization of D2 autoreceptor-dependent GIRK currents in wild type dopamine neurons, but promoted desensitization of D2S receptors.

In wild type neurons with EGTA internal, isradipine had no significant effect on the magnitude of the maximal current produced by bath application of quinpirole (control: $172 \pm 19$ pA, $n = 15$; israd: $178 \pm 28$ pA, $n = 11$, $p = 0.86$, unpaired $t$ test). Therefore, the effect of isradipine on I-DA was examined. In wild type dopamine neurons using an EGTA internal, isradipine (300 nM) significantly augmented I-DA (*Figure 6D*) after >10 min (*Figure 6—figure supplement 1*). In D2S and D2L neurons, I-DA was also augmented by isradipine (*Figure 6E* and *Figure 6—figure supplement 1*).

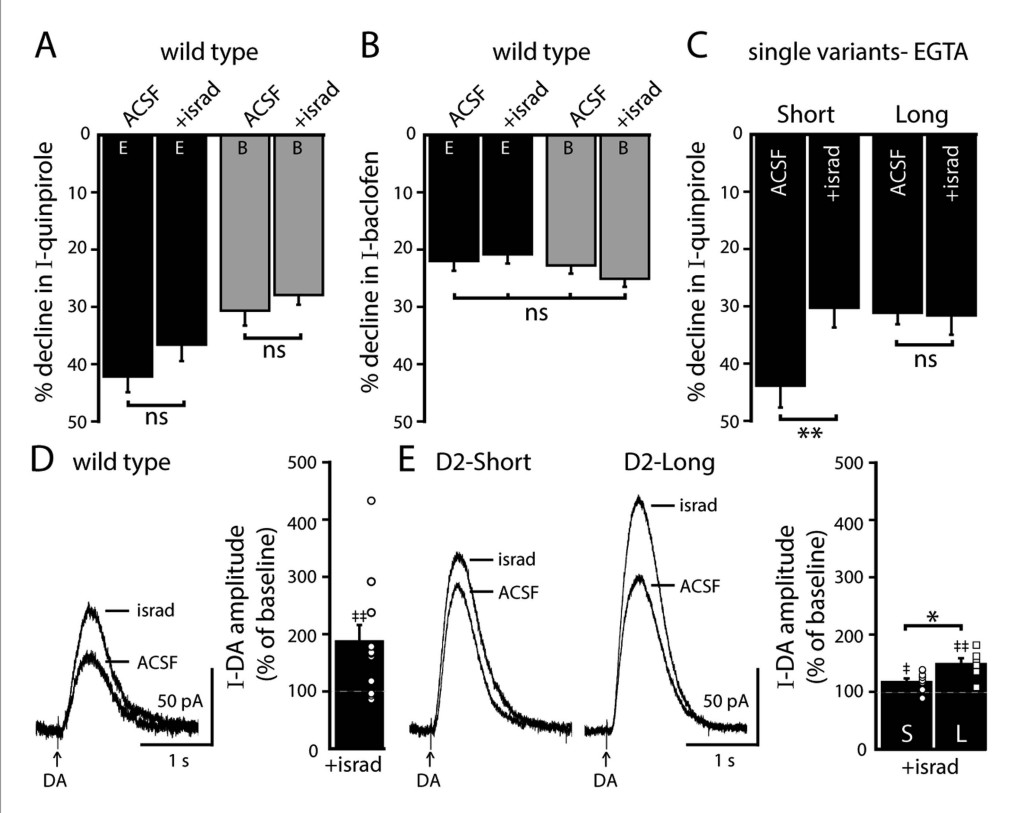

**Figure 6**. Blocking L-type calcium channels differentially modifies D2S and D2L receptor-dependent GIRK conductance. (**A**, **B**) In wild type neurons, isradipine (300 nM, > 20 min) had no significant effect on the decline in quinpirole-induced (**A**) and baclofen-induced current (**B**) recorded with EGTA and BAPTA internal (quinpirole: $n = 11–15$, one-way ANOVA followed by Fisher's LSD; baclofen: $n = 12–16$, one-way ANOVA). (**C**) In D2S, but not D2L neurons, isradipine reduced the decline in quinpirole-induced current using EGTA internal ($n = 6–16$, one-way ANOVA followed by Fisher's LSD). (**D**, **E**) Submaximal D2 receptor-dependent outward currents were produced by iontophoretic application of dopamine once every 50 s while recording with EGTA internal (I-DA, arrows). (**D**) Isradipine (300 nM, > 15 min) augmented I-DA in wild type neurons, shown in representative averaged traces (left) and grouped data (right, $n = 11$). (**E**) Isradipine augmented I-DA in D2S and D2L neurons, shown in representative averaged traces (left) and grouped data (right). The augmentation by isradipine was greater in D2L than D2S neurons ($n = 6–11$, unpaired $t$ test). The time course of the isradipine-induced augmentation of I-DA can be found in *Figure 6—figure supplement 1*. Baseline: mean amplitude of six I-DAs preceding isradipine application, ns indicates not significant, *p < 0.05, **p < 0.01, and ‡ indicates significance over baseline (within-group comparison, paired $t$ tests).

The following figure supplement is available for figure 6:

**Figure supplement 1**. Prolonged isradipine application enhances D2 receptor-dependent currents produced by exogenous dopamine.

As was found with CPA, the magnitude of the increase was greater for D2L receptor-dependent currents than D2S (*Figure 6E*). Thus, inhibition of calcium entry via L-type calcium channels differentially increased D2S and D2L receptor-dependent GIRK signaling.

## A single in vivo cocaine exposure decreases calcium-dependent D2 autoreceptor desensitization

Drugs of abuse change the D2 autoreceptor activation of GIRK conductance (*Arora et al., 2011*; *Gantz et al., 2013*; *Dragicevic et al., 2014*; *Sharpe et al., 2014*). One of these changes is a loss of the calcium-dependent component of D2 autoreceptor desensitization after repeated ethanol exposure

(*Perra et al., 2011*). Wild type mice were given a single injection of cocaine (20 mg/kg, i.p.) or an equal volume of saline, and brain slices were made 24 hr later. With EGTA internal, the quinpirole-induced current declined significantly less in slices from cocaine-treated mice compared to control mice (saline-treated and naïve, *Figure 7A*). In fact, the decline was no longer statistically different from that found with BAPTA internal (*Figure 7A*). There was no difference in the decline in the quinpirole-induced current in slices taken from control or cocaine-treated mice with BAPTA internal (*Figure 7A*). There was no change in the decline in the baclofen-induced current, using either internal (*Figure 7F*). Thus, in wild type mice, a single in vivo cocaine exposure resulted in the loss of calcium-dependent D2 autoreceptor desensitization without affecting GABA_B receptor desensitization.

A loss of calcium-dependent D2 autoreceptor desensitization after cocaine exposure in wild type neurons could be due to a change in calcium signaling or a functional increase in the contribution of

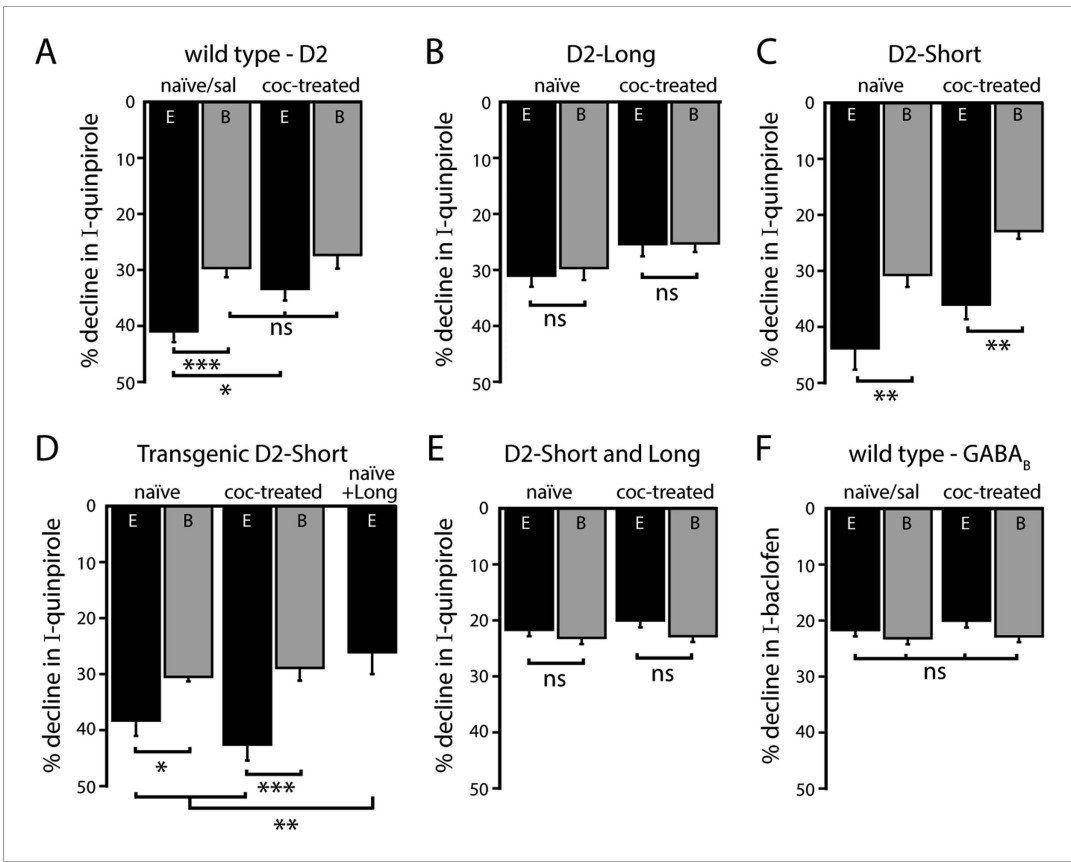

**Figure 7**. Effects of a single in vivo cocaine exposure on calcium-dependent D2 autoreceptor desensitization. (**A**) In neurons from cocaine-treated wild type mice using EGTA internal, the quinpirole-induced current declined less compared to naïve/saline-treated mice, to a level comparable to the decline recorded with BAPTA internal. Cocaine exposure did not alter the decline in the quinpirole-induced current when measured with BAPTA internal ($n$ = 11–26). (**B**) In D2L neurons after in vivo cocaine exposure, there was no difference in the decline in quinpirole-induced current recorded with EGTA or BAPTA internal ($n$ = 6–7). (**C, D**) In neurons from (**C**) AAV-D2S and (**D**) transgenic D2S mice, the decline in quinpirole-induced current was still greater using EGTA internal compared to BAPTA after in vivo cocaine exposure (**C**: $n$ = 10 each, **D**: $n$ = 8–9). (**D, E**) Co-expression of both splice variants by (**D**) viral expression of D2L in transgenic D2S mice and (**E**) infection with a mixture of AAV-D2S and AAV-D2L removed the calcium-dependence of the decline in the quinpirole-induced current (**D**: $n$ = 5–8, **E**: $n$ = 6–9) and there was no change after in vivo cocaine (**E**: $n$ = 7–9). (**F**) Previous cocaine exposure had no effect on the decline in baclofen-induced current recorded with EGTA or BAPTA internal in wild type neurons ($n$ = 13–27). Comparisons were made with one-way ANOVAs followed when $p < 0.05$ by Fisher's LSD. ns indicates not significant, *$p < 0.05$, **$p < 0.01$, ***$p < 0.001$.

D2L receptors. To test whether D2L receptors are involved, $Drd2^{-/-}$ mice that had received midbrain injections of AAV-D2S or AAV-D2L were given a single injection of cocaine (20 mg/kg, i.p.) and brain slices were made 24 hr later. In D2L neurons, cocaine exposure did not alter the decline in quinpirole-induced current (*Figure 7B*). Likewise, cocaine exposure did not alter the calcium-dependent decline in the quinpirole-induced current in D2S neurons. Similar to naïve AAV-D2S mice, the decline of the quinpirole-induced current was greater using EGTA internal than with BAPTA internal (*Figure 7C*). Thus, unlike what was found in slices from wild type mice, cocaine exposure did not reduce the calcium-dependent decline in the quinpirole-induced current. This result was not dependent on overexpression as it was also observed in D2S neurons in which quinpirole produced outward currents of similar magnitude to wild type neurons (data not shown). This result was also recapitulated in the transgenic D2S mice (*Figure 7D*), where expression of D2S receptors in the midbrain is restricted to dopamine neurons. Since wild type, but not D2S-only dopamine neurons exhibited a reduction in calcium-dependent desensitization after cocaine exposure, these results suggest that constitutive or viral-mediated expression of D2S as the exclusive autoreceptor was insufficient for cocaine-induced plasticity.

To determine if the expression of D2L was sufficient to enable loss of calcium-dependent D2 receptor desensitization of D2S, $Drd2^{-/-}$ mice received bilateral injections of a 1:1 mixture of AAV-D2S and AAV-D2L. In dopamine neurons from mice infected with both splice variants, the amplitude of the quinpirole-induced currents was similar to those measured in neurons expressing D2S- or D2L-only (EGTA: 333 ± 48 pA, $n = 9$; BAPTA: 442 ± 86 pA, $n = 6$, see *Figures 1B, 4C* for comparison). This suggests a similar level of D2 receptor overexpression as in neurons that express single variants. Surprisingly, the decline in the quinpirole-induced current was similar between EGTA and BAPTA internals (*Figure 7E*). Therefore, the viral expression of both D2S and D2L receptors did not mimic D2 receptor-dependent GIRK signaling in naïve wild type mice. Moreover, the decline in the quinpirole-induced current did not change after in vivo cocaine exposure (*Figure 7E*), suggesting that the viral expression of both D2S and D2L receptors precluded cocaine-induced plasticity in D2 receptor-dependent GIRK signaling. To ensure that this result was not due to preferential expression of D2L receptors following injection of the D2S/D2L virus mixture, dopamine neurons in transgenic D2S mice were infected with AAV-D2L. The expression of D2S was confirmed by labeling dopamine neurons with an Alexa Fluor 594-conjugated M1 antibody and imaging on a two-photon microscope (e.g., *Figure 4—figure supplement 2B*). Recordings were made from neurons with Flag-D2S staining that were also GFP+ (AAV-D2L). With EGTA internal, the decline in the quinpirole-induced current in transgenic D2S neurons also expressing D2L was equivalent to the decline measured in neurons receiving the D2S/D2L virus mixture, and significantly less than the decline in the quinpirole-induced current in transgenic D2S-only neurons (*Figure 7D*). As observed in wild type, there was no change in the decline in the baclofen-induced current after cocaine exposure in any of the groups ($p > 0.05$ for all comparisons, data not shown). Taken together, the results indicate that regardless of the presence of D2S, the viral expression of D2L eliminated calcium-dependent D2 receptor desensitization and precluded cocaine-induced plasticity.

## Discussion

Alternative splicing generates two isoforms of the dopamine D2 receptor, D2S and D2L. D2S has been considered the autoreceptor, but both are expressed and functional in midbrain dopamine neurons. Evidence of distinct functional roles for the splice variants as autoreceptors has not been described. This study assessed the calcium-dependence and drug-induced plasticity of the desensitization of GIRK currents mediated by D2S and D2L receptors expressed in SNc dopamine neurons. The results reveal that the D2S receptor, but not the D2L receptor, exhibited calcium-dependent desensitization. Manipulating pathways for calcium signaling removed the calcium-dependent component of D2S receptor desensitization, demonstrating these receptors were amenable to plasticity. Cocaine exposure eliminated calcium-dependent D2 autoreceptor desensitization in dopamine neurons from wild type mice without altering desensitization of neurons expressing only D2S or D2L. Furthermore, viral expression of D2L eliminated calcium-dependent desensitization, resembling the D2 autoreceptor desensitization observed in dopamine neurons from wild type mice after in vivo cocaine exposure.

## Calcium-dependent regulation of D2 autoreceptor-dependent GIRK conductance

Calcium entry promotes desensitization of D2 autoreceptors in wild type dopamine neurons. Buffering intracellular calcium with BAPTA reduces the decline in D2 autoreceptor-, but not GABA$_B$ receptor-mediated GIRK currents (*Beckstead and Williams, 2007*; *Perra et al., 2011*). In this study, the calcium-dependent component of D2 autoreceptor desensitization was observed in wild type and D2S-only expressing dopamine neurons. However, in neurons where D2L receptors were virally expressed, there was no calcium-dependent desensitization. This was observed whether D2L receptors were expressed alone, or in conjunction with transgenic or virally expressed D2S receptors.

This study describes two calcium sources that regulate D2 autoreceptor-dependent GIRK currents: intracellular calcium stores and L-type calcium channels. These intracellular pathways did not regulate desensitization of GABA$_B$ receptor-dependent GIRK currents. Consistent with a previous report (*Perra et al., 2011*), depleting intracellular calcium stores removed calcium-dependent D2 autoreceptor desensitization in wild type neurons. Depleting intracellular calcium stores also reduced the magnitude of D2S receptor desensitization to a saturating concentration of agonist, without affecting D2L receptor desensitization. Preventing calcium entry from L-type calcium channels also reduced D2S receptor desensitization, without affecting wild type or D2L receptor desensitization. The results demonstrate that the calcium-dependent component of D2S receptor desensitization was readily modifiable.

Desensitization of D2 autoreceptors in wild type dopamine neurons was controlled by elevated concentrations of calcium in intracellular microdomains and could not be enhanced by raising the resting free calcium concentration. The lack of a calcium-dependent component in D2L receptor desensitization could be due to localization outside of the calcium microdomains, despite showing similar distribution in the somatodendritic compartment as D2S receptors (*Jomphe et al., 2006*), or to another property of this isoform. Depleting intracellular calcium stores or blocking L-type calcium channels produced robust augmentation in D2L receptor-dependent GIRK currents produced by iontophoretically applied dopamine that was greater than the augmentation of D2S receptor-dependent GIRK currents. The lack of effect of manipulating calcium signaling on D2L receptor desensitization in the presence of a saturating concentration of quinpirole suggests that the enhanced response of D2L to iontophoretically applied dopamine does not reflect removal of tonic desensitization. Nonetheless, the same intracellular pathways interacting with D2S receptors also modified D2L receptor-dependent GIRK currents. It is therefore likely that D2S and D2L receptors are in similar calcium microdomains and the lack of apparent calcium-dependent desensitization upon saturating agonist exposure is a property specific to the D2L isoform.

## Plasticity of the calcium-dependent D2 autoreceptor desensitization

Drugs of abuse cause functional changes to dopamine neuron physiology, including regulation of D2 and GABA$_B$ receptor activation of GIRK conductance (*Beckstead et al., 2009*; *Arora et al., 2011*; *Perra et al., 2011*; *Padgett et al., 2012*; *Dragicevic et al., 2014*; *Sharpe et al., 2014*). Several recent studies reported drug-induced changes in D2 autoreceptor mediated-GIRK signaling that are contingent on the method of recording (whole-cell vs perforated-patch, *Dragicevic et al., 2014*) or the calcium buffering capabilities of the whole-cell internal solution (*Perra et al., 2011*; *Sharpe et al., 2014*) implicating dependence on intracellular calcium. In this study, 24 hr after a single in vivo cocaine exposure, the calcium-dependent component of D2 autoreceptor desensitization was eliminated, similar to the change observed after repeated ethanol exposure (*Perra et al., 2011*). Thus, this study confirms the *Perra et al. (2011)* finding and further demonstrates that the plasticity in D2 autoreceptor function did not require repeated drug exposure. Whether this plasticity was due to a change in calcium-dependent pathways or the D2 autoreceptors themselves was previously unresolved. The findings of this study support the latter.

Cocaine exposure may change the calcium-dependent pathways examined in this study. Twenty-four h after a single cocaine exposure, metabotropic glutamate receptor 1 signaling is attenuated (*Kramer and Williams, 2015*). The activation of metabotropic glutamate receptors decreases D2 autoreceptor-dependent GIRK currents (*Perra et al., 2011*) and may desensitize D2 autoreceptors through calcium release from intracellular stores. A change in the contribution of calcium influx via L-type calcium channels to D2 autoreceptor desensitization may also result from cocaine exposure (*Dragicevic et al., 2014*). In this study, depleting intracellular calcium

stores or blocking L-type calcium channels readily removed calcium-dependent D2S receptor desensitization. Given these results, it was surprising that cocaine exposure did not alter calcium-dependent D2S receptor desensitization. This result was recapitulated in dopamine neurons from transgenic D2S mice indicating it was not an artifact of virus-mediated expression. Thus, these results suggest that the expression of D2S as the exclusive autoreceptor is insufficient for cocaine-induced plasticity observed in wild type dopamine neurons.

In wild type dopamine neurons, it may be that the expression of D2L receptors is involved in cocaine-induced plasticity. Biased expression of D2S and D2L receptors has been associated with drug abuse. The loss of D2L receptors and concomitant overexpression of D2S receptors in D2L-deficient mice is associated with altered drug-taking (*Bulwa et al., 2011*) and conditioned place preference (*Smith et al., 2002*). In addition, single nucleotide polymorphisms that result in overexpression of D2S receptors are observed in humans with a history of drug abuse (*Sasabe et al., 2007*; *Moyer et al., 2011*). In this study, the viral expression of D2L receptors, alone or with D2S receptors, resulted in a loss of calcium-dependent D2 receptor desensitization. Moreover, it precluded any further cocaine-induced reduction in calcium-dependent D2 receptor desensitization. These results suggest that the overexpression of D2L receptors resembles cocaine-induced plasticity. Transient elevation in extracellular dopamine up-regulates D2L mRNA (*Zhang et al., 1994*; *Oomizu et al., 2003*; *Giordano et al., 2006*; *Wernicke et al., 2010*; but see; *Filtz et al., 1993*; *Dragicevic et al., 2014*). In addition, D2L receptors are retained in intracellular compartments more so than D2S receptors and exposure to D2 agonists results in the preferential translocation of existing and nascent D2L receptors to the membrane (*Filtz et al., 1993*; *Zhang et al., 1994*; *Fishburn et al., 1995*; *Starr et al., 1995*; *Ng et al., 1997*; *Prou et al., 2001*). Thus, it may be that exposure to cocaine in wild type mice increases functional D2L receptors, resulting in the loss of calcium-dependent D2 autoreceptor desensitization. Virally expressed D2L receptors may not be subject to the same regulation as endogenously expressed D2L receptors, in such a way that virus-mediated overexpression of D2L mimics and eliminates any requirement for up-regulation of D2L function.

## Both D2S and D2L function as autoreceptors

The D2S isoform has been thought to be the D2 autoreceptor due to preservation of autoreceptor-mediated behaviors in D2L-deficient mice and more abundant D2S immunolabeling in the SNc (*Khan et al., 1998*; *Usiello et al., 2000*). However, immunolabeled D2L receptors are found in SNc dopamine neurons (*Khan et al., 1998*) and rodent studies describe dopamine neurons expressing both D2S and D2L mRNA, D2L-only, or D2S-only (*Jang et al., 2011*; *Dragicevic et al., 2014*). Both variants are capable of inhibiting action potential firing (*Jomphe et al., 2006*; *Jang et al., 2011*; *Dragicevic et al., 2014*). In this study, D2S and D2L receptors, when expressed in dopamine neurons, activated a GIRK conductance and were capable of producing IPSCs occurring from spontaneous fusion of dopamine-filled vesicles. Thus, D2S and D2L can serve as autoreceptors at somatodendritic dopamine synapses, as previously demonstrated (*Neve et al., 2013*).

Although many of the basic properties of D2S and D2L receptor-dependent currents were similar, there were some differences that suggest both D2S and D2L are autoreceptors in wild type dopamine neurons. The calcium-dependent component of D2 autoreceptor desensitization in wild type neurons was similar to D2S-only neurons. However, results from manipulating calcium signaling in wild type neurons were more consistent with a mix of D2S and D2L receptor expression. Intracellular pathways for calcium signaling that regulated D2S receptor-GIRK desensitization modified the magnitude of D2L receptor-dependent GIRK currents. In wild type dopamine neurons, manipulating these pathways resulted in a decrease in acute desensitization and larger GIRK currents, suggesting that D2S and D2L receptor regulation may operate in parallel in wild type neurons. In addition, cocaine-induced plasticity occurred in wild type, but not D2S-only neurons, indicating a loss of some process in neurons which express D2S as the exclusive autoreceptor that is permissive to cocaine-induced plasticity. However, the viral-mediated co-expression of D2S with D2L receptors also did not resemble wild type, and instead was similar to D2L-only. Although it is not known to what extent developmental compensation, virus-mediated expression, and variegated D2 receptor expression in *Drd2*$^{-/-}$ mice affected D2 receptor translation or trafficking (i.e. affecting the ratio of functional D2S and D2L receptors), or other regulatory elements of D2 receptor signaling, this result suggests that in wild type

dopamine neurons, the functional expression of D2L may be limited. Changes in calcium signaling or exposure to cocaine may bring about an increased contribution of D2L, although this has yet to be directly demonstrated. Taken together, this study suggests that D2S may serve as the predominant autoreceptor under basal conditions, but the functional contribution of D2L autoreceptors may be revealed after drug exposure.

## Concluding remarks

This study advances the understanding of D2 autoreceptor regulation. Two pathways for calcium signaling that regulated D2 autoreceptor-dependent GIRK signaling were identified, which distinctly affected D2S and D2L receptors. In addition, distinct action of in vivo cocaine exposure in wild type, D2S, and D2L receptor-GIRK signaling was demonstrated. Since not all dopamine neurons express both D2S and D2L receptors, this study suggests that D2 autoreceptors in a subset of dopamine neurons are regulated differently by calcium and resistant to cocaine-induced plasticity. Given the heterogeneity of dopamine neurons and their projections (reviewed in *Roeper, 2013*), a greater understanding of this subset may reveal insights into plasticity in their projection areas.

## Materials and methods

### Animals

All studies were conducted in accordance with the Institutional Animal Care and Use Committees at the VA Portland Health Care System (VAPORHCS) and Oregon Health & Science University (OHSU). Mice of both sexes were used in this study (65–120 days old). Wild type (C57BL/6) mice, obtained from The Jackson laboratory (Sacramento, CA), and TH-hD2S mice were bred at OHSU. $Drd2^{-/-}$ mice were bred at the VAPORHCS Veterinary Medical Unit and were maintained on a C57BL/6 background. Mice were housed in standard plastic containers on a 12 hr light/dark cycle. Food and water were available *ad libitum*, and after stereotaxic injections, diet was supplemented with Diet Gel RE placed on the floor of the cage. 'Transgenic D2S' mice were produced by crossing $Drd2^{-/-}$ mice with transgenic TH-hD2S mice, which express Flag-tagged human D2S receptors under the tyrosine hydroxylase promoter (*Gantz et al., 2013*), as shown by immunostaining for the Flag epitope with an Alexa Fluor 594-conjugated M1 antibody and confocal or two-photon microscopy (*Figure 4—supplement 2*). Treated animals received one injection of cocaine (20 mg/kg, intraperitoneally) dissolved in saline, or an equal volume of saline, 22–24 hr prior to use. There were no differences found between saline-treated and naïve mice, so data were combined.

### Stereotaxic injections and viruses

D2 receptors were ubiquitously expressed in the midbrain using an AAV vector (AAV9 D2-IRES-GFP; Virovek, Inc., Hayward, CA) encoding rat D2S or D2L receptors (*Neve et al., 2013*), or a 1:1 mixture of AAV-D2S and AAV-D2L. Mice were injected when 65–90 days old. Mice were immobilized in a stereotaxic alignment system under an anesthesia cocktail consisting of 7.1 mg/kg xylazine, 71.4 mg/kg ketamine, and 1.4 mg/kg acepromazine (10 ml/kg). Mice received bilateral injections, each 500 nl volume at a rate of 200 nl/min, with the injection needle left in place for an additional 5 min before it was slowly withdrawn. The coordinates for injections were (with respect to bregma) AP −3.26 mm, ML ±1.2 mm, DV −4.0 mm. After injections, mice recovered in individual or group housing for 3–4 weeks to allow for expression. Infected neurons were identified in the slice by visualization of eGFP.

### Slice preparation and electrophysiology

Whole-cell voltage clamp recordings (holding potential −60 mV) were made as previously described (*Gantz et al., 2013*). Whenever possible, experiments were conducted blinded to treatment or splice variant expression. Mice were deeply anesthetized with isoflurane and killed by decapitation. Brains were removed quickly and placed in ice-cold physiologically equivalent saline solution (modified Krebs buffer) containing (in mM) 126 NaCl, 2.5 KCl, 1.2 $MgCl_2$, 2.4 $CaCl_2$, 1.4 $NaH_2PO_4$, 25 $NaHCO_3$, and 11 D-glucose with MK-801 (10 μM), and cut horizontally (220 μm) using a vibrating microtome (Leica). Slices recovered at 30°C in vials with 95/5% $O_2/CO_2$ saline with MK-801 (10 μM) for at least 30 min prior to recording. Slices were then mounted in a recording chamber and perfused at a rate of ~3.0 ml/min with 33–35.5°C modified Krebs buffer. Recordings were made exclusively from neurons in

the SNc identified visually by their location lateral to the medial terminal nucleus of the accessory optic. The neurochemical identity of AAV-D2-infected cells was not verified post-recording. Rather, dopamine neurons were identified by location and electrophysiological properties, namely the presence of spontaneous pacemaker firing of broad (~2 ms) action potentials at 1–5 Hz in cell-attached mode (*Ford et al., 2006*), characteristic passive membrane properties including capacitance and resting conductance (*Gantz et al., 2011*), and a prominent slow hyperpolarization-activated inward ($I_h$) current (*Ford et al., 2006*). These parameters readily distinguished dopamine neurons from GABAergic neurons. The expression of D2 receptors was not used as a physiological criterion for dopamine neuron identity. However, all dopamine neurons from wild type mice identified by location and electrophysiological properties had a D2 receptor-dependent outward current upon quinpirole application. Recordings were obtained with large glass electrodes with a resistance of 1.3–1.9 M$\Omega$ when filled with internal solution containing either, (in mM) 115 K-methanesulfonate, 20 NaCl, 1.5 $MgCl_2$, 10 HEPES (K), 2 ATP, 0.2 GTP, 10 phosphocreatine, and 10 BAPTA (K4) or 130 K-methanesulfonate, 20 KCl, 1 $MgCl_2$, 10 HEPES (K), 2 ATP, 0.2 GTP, 10 phosphocreatine, and 0.1 EGTA; pH 7.33–7.40, 275–288 mOsm. The concentration of $CaCl_2$ required to increase resting free calcium in BAPTA internal was determined with use of the EGTAetc program, provided by EW McCleskey. Within 2 min of break-in, membrane capacitance, series resistance, and input resistance were measured with the application of 3 pulses ($\pm$2 mV for 50 ms) averaged before computation using the Axograph (sampled at 50 kHz, filtered at 10 kHz). Series resistance was monitored to ensure sufficient and stable electrical access to the inside of the cell throughout the experiment (<12 M$\Omega$). Cells were dialyzed with internal solution for >10 min prior to drug application (*Foehring et al., 2009*). All drugs were applied through bath perfusion, except dopamine, which was applied by iontophoresis. Quinpirole and baclofen were applied with >10 min between the agonist applications with the application order alternated between recordings. The amplitude and the decline in the currents were not affected by the order in which the drugs applied. Slices were exposed to saturating concentrations of each agonist once. Recordings where the peak amplitude of the current was <50 pA were excluded from decline analysis. Dopamine hydrochloride (1 M) was ejected as a cation with a single pulse (2–10 ms, >20 nA) from a thin-walled iontophoretic electrode placed within 10 µm of the soma once every 50 s. Access resistance was assessed during these recordings with a brief (200 ms) step to −70 mV once every 50 s. Data were acquired using AxoGraph software (AxographX, Berkeley, CA) and Chart 7 (AD Instruments, Colorado Springs, CO) and were post hoc filtered. The amplitude of currents induced by iontophoretic application of dopamine was determined by averaging the current $\pm$20 ms from the greatest upward deflection. For each cell, 6–24 consecutive currents were averaged to determine 'baseline' (preceding drug application) and post-drug amplitudes. sIPSCs were detected and analyzed as previously described (*Gantz et al., 2013*). Briefly, single peak sIPSCs with amplitudes greater than 2.1 times the SD of baseline noise were detected using a semiautomated sliding template detection procedure with AxoGraph X. Each detected event was visually inspected and discarded if the baseline noise was greater than the sIPSC peak $\pm$1 s from the peak. Duration of sIPSCs was determined by measuring the width at 20% of the peak.

## Flag-D2S receptor immunohistochemistry and microscopy

Brain slices were prepared and allowed to recover, as described for electrophysiology. Slices were incubated in Alexa Fluor 594-conjugated M1 antibody (10 µg/ml) for 40 min at 35°C. Live slices were observed with a custom-built two-photon microscope using *ScanImage* Software (*Pologruto et al., 2003*). Expression of eGFP was visualized using a CCD camera of epi-fluorescence activation. Slices for laser-scanning confocal microscopy were washed 10 min in modified Krebs buffer before fixation in 4% paraformaldehyde (45 min at 24°C) in phosphate buffered saline + $CaCl_2$ (1 mM, PBS+$Ca^{2+}$). Slices were blocked and permeabilized in PBS+$Ca^{2+}$ with 0.3% Triton-X and 0.5% fish skin gelatin for 80 min. Slices were incubated overnight in rabbit anti-tyrosine hydroxylase antibody (1:1000 in PBS+$Ca^{2+}$ + 0.05% $NaN_3$). Washed slices were incubated in Alexa Fluor 488-conjugated goat anti-rabbit secondary antibody (1:1000 in PBS+$Ca^{2+}$ + 0.05% $NaN_3$, 2 hr at 24°C). Washed slices were mounted with Fluoromount aqueous medium with #1.5 glass coverslips. Images were collected on a Zeiss confocal LSM 780 microscope with a 40× water-emersion lens (1.2 nA). All images were processed with Fiji.

## Materials

CGP-55845 was obtained from Tocris Bioscience (Minneapolis, MN). MK-801 and CPA were obtained from Abcam (Cambridge, MA). Cocaine hydrochloride was obtained from National Institute on Drug Abuse-National Institutes of Health (Bethesda, MD). All other drugs were obtained from Sigma–Aldrich (St. Louis, MO).

## Statistical analyses

Values are given as means $\pm$ SEM and unless otherwise noted $n =$ number of cells. Data sets with $n > 10$ were tested for normality with a Shapiro–Wilk test. Significant between-group differences were determined in two group comparisons by unpaired two-tailed $t$ tests or Mann–Whitney $U$ tests, and in more than two groups comparisons by one- or two-way ANOVAs. ANOVAs were followed when $p < 0.05$ with uncorrected Fisher's LSD or Bonferroni's multiple comparisons post hoc tests. Significant differences in within-group comparisons were determined by paired two-tailed $t$ tests. Statistical analysis was performed using GraphPad Prism 6 (GraphPad Software, Inc., La Jolla, CA).

## Acknowledgements

This work was supported by Merit Review Award BX000810 from the US Department of Veterans Affairs, Veterans Health Administration, Office of Research and Development, Biomedical Laboratory Research and Development (KAN) and by NIH P50 DA018165 (KAN), F32 DA038456 (BGR), DA04523 (JTW), and DA034388 (JTW).

## Additional information

### Funding

| Funder | Grant reference | Author |
| --- | --- | --- |
| National Institutes of Health | P50 DA018165 | Kim A Neve |
| U.S. Department of Veterans Affairs | BX000810 | Kim A Neve |
| National Institutes of Health | R01 DA004523 | John T Williams |
| National Institutes of Health | R01 DA034388 | John T Williams |
| National Institutes of Health | F32 DA038456 | Brooks G Robinson |

The funders had no role in study design, data collection and interpretation, or the decision to submit the work for publication.

### Author contributions

SCG, BGR, Conception and design, Acquisition of data, Analysis and interpretation of data, Drafting or revising the article; DCB, Conception and design, Acquisition of data; JRB, RLN, Conception and design, Contributed unpublished essential data or reagents; JTW, KAN, Conception and design, Analysis and interpretation of data, Drafting or revising the article

### Author ORCIDs

Stephanie C Gantz, http://orcid.org/0000-0002-1800-4400

### Ethics

Animal experimentation: All studies were conducted in accordance with the recommendations in the Guide for the Care and Use of Laboratory Animals of the National Institutes of Health and were approved by the Institutional Animal Care and Use Committees at the VA Portland Health Care System (#2577-12) and Oregon Health & Science University (IS01394).

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
