## [Decision Letter]

Thank you for submitting your work entitled “Distinct regulation of dopamine D2S and D2L autoreceptor signaling by calcium” for peer review at *eLife*. Your submission has been favorably evaluated by Eve Marder (Senior Editor) and four reviewers, one of whom, Sacha Nelson, is a member of our Board of Reviewing Editors.

The reviewers have discussed the reviews with one another and the Reviewing Editor has drafted this decision to help you prepare a revised submission.

In this elegant study the authors use knockout mice and viral expression of specific isoforms of the D2 receptor to express one or both of the normally expressed isoforms to show that, contrary to earlier thoughts, both short and long isoforms function as autoreceptors. The results also reveal selective roles of the two subtypes in mediating desensitization of the autoreceptor response.

Essential revisions:

All three reviewers felt that the roles of D2S and D2L in cocaine-induced plasticity are unclear. For example, one reviewer points out: “The Abstract states: ‘expression of D2S as the exclusive autoreceptor was insufficient for cocaine-induced plasticity, demonstrating a functional role for the co-expression of D2S and D2L auto receptors.’ However, the Results state: ‘regardless of the presence of D2S, the expression of D2L eliminated calcium-dependent D2 receptor desensitization and precluded cocaine-induced plasticity’. If some DA neurons express either D2S or D2L, they do not show cocaine-induced autoreceptor plasticity, and if D2L blocks plasticity, then it is not clear under what conditions cocaine would induce autoreceptor plasticity (in WT mice).” Another reviewer suggested: “The authors should discuss why this might be the case. For example is it possible that cocaine treatment upregulates D2L in wild type neurons, thus eliminating calcium-dependent desensitization? Could this be tested? Or alternatively, perhaps the cellular location of D2Rs expressed after viral infection is sufficiently distant from the sites of cocaine action to preclude the drug effect. The spontaneous IPSC data suggest that receptors can be located near sites of DA release, but perhaps the majority of them are not so close. Other factors that could affect this outcome should also be considered.” Although additional tests are suggested, this point could be adequately addressed with further discussion.

A further point of suggested additional discussion concerns the interpretation of the fact that treatments that potentiate D2-activated GIRK current had little effect on D2S expressing neurons. Could D2L receptors be desensitized by endogenous DA to a greater extent than D2S receptors, and could this long-lasting desensitization be reversed by suppressing calcium signaling?

The reviewers also felt that it was important to acknowledge a few caveats. First, presumably cells varied in their degree of transfection and some cells may have expressed receptors at supra physiologic levels. Second, knockout could have produced some compensation. Finally, although it was likely most of the neurons patched were DA neurons, it would be important to acknowledge that the neurochemical identity of the neurons was not verified.

A couple of other clarifications or amplifications were requested: Were subtype specific effects observed in all cell recorded? How reliable was the cocaine-induced behavioral response and was it accompanied by hyperlocomotion? It was not clear to one reviewer how L channels were activated in the experiments in Figure 6 and the accompanying supplement. Was the membrane potential held at -60 mV throughout? Do the authors think the relevant portion of the cell escaped voltage clamp? Another reviewer was confused about why the authors start using ionotophoretically-ejected DA in Figure 5, but not in the previous figures. Why not more systematically compare saturating and subsaturating D2R activation across the whole paper? Also, do the authors have evidence that the DA-mediated currents are completely D2R-dependent? This reviewer also felt that justification of the high dose of quinpirole was called for. One reviewer suggested: “It would be nice to see stimulus evoked D2/GIRK-mediated IPSCs in addition to the spontaneous synaptic responses shown in Figure 1. This might indicate if the synaptically-accessible D2 pool is similar in the infected neurons.”

Although the authors may have some of the additional requested data or be able to obtain them quickly and this might improve the manuscript, addressing the points that can be addressed through textual clarification should be sufficient for acceptance.

Finally, one reviewer had major suggestions for improving the presentation which the other reviewers agreed would be beneficial: “The work would be more convincing and easy to follow if the results of experiments with WT DA neurons was compared, in each figure, with the results obtained with neurons expressing D2S only, D2L only or the combination of both. This would help to make clearer how individual isoforms recapitulate the observations made in neurons expressing the native receptors. Such a comparison would be more informative than the comparison performed with the GABAB agonist, which is interesting, but perhaps not as informative.”

[Editors' note: further revisions were requested prior to acceptance, as described below.]

Thank you for submitting your work entitled “Distinct regulation of dopamine D2S and D2L autoreceptor signaling by calcium” for peer review at *eLife*. Your submission has been favorably evaluated by Eve Marder (Senior Editor) and three reviewers, one of whom is a member of our Board of Reviewing Editors.

The reviewers have discussed the reviews with one another and the Reviewing Editor has drafted this decision to help you prepare a revised submission.

The following individuals responsible for the peer review of your submission have agreed to reveal their identity: Sacha Nelson (Reviewing Editor and peer reviewer) and David Loving (peer reviewer).

As you can see, one of the reviewers still feels that the conclusion about the role of the D2L in the cocaine-dependent plasticity needs to be toned down and that additional caveats should be discussed. Specifically the reviewer felt that because of the viral strategy used, the possibility of non-cell autonomous effects (e.g. of expressing D2R in GABAergic neurons) could not be ruled out. They also felt that the conclusion about D2L is over stretched, since the experiments do not directly demonstrate the role (restoring D2S is not enough, nor is D2 + D2L). Discussion between the Reviewing Editor and the other two reviewers led to agreement that textual changes in the Discussion to acknowledge these caveats would be sufficient.

Reviewer #1:

It appears that the authors have suitably responded to the many points raised by the other reviewers.

Reviewer #2:

The study convincingly demonstrates the role of D2S receptors in calcium-sensitive desensitization and elaborates on two mechanisms.

However, the argument that restoring D2S expression in D2R KO mice does not confer WT cocaine-induced plasticity, and that it therefore involves D2L receptors is weak, given that viral restoration of D2S and D2L receptors together does not confer the plasticity. There are other explanations for why restoration of D2L does not restore the plasticity, many of which are discussed, but still do not strengthen the conclusion that D2L autoreceptors are required for cocaine-induced plasticity.

Some of the confounds in the interpretation of the results may have to do with lack of specificity of the D2S and D2L restoration to dopamine neurons (as all cells in the injection site were non-specifically transfected). It would have been cleaner to use D2R KO mice also carrying the DAT^IREScre^ allele and conditional viruses to restrict D2R subtype expression to dopamine neurons. Furthermore, the identification of dopamine neurons by physiological criteria is potentially impacted by D2R expression, and may have confounded the results.

Reviewer #3:

As previously stated, this study is convincing, and provides intriguing new data about differential desensitization mechanisms and in vivo cocaine effects on the D2 splice variants. The quality of the data is high, and the experimental strategies are innovative.

The authors have adequately addressed all of the concerns raised in the initial review, and the story is no more focused and complete, with appropriate discussion.

---

## [Author Response]

*All three reviewers felt that the roles of D2S and D2L in cocaine-induced plasticity are unclear. For example, one reviewer points out: “The Abstract states: ‘expression of D2S as the exclusive autoreceptor was insufficient for cocaine-induced plasticity, demonstrating a functional role for the co-expression of D2S and D2L auto receptors.’ However, the Results state: ‘regardless of the presence of D2S, the expression of D2L eliminated calcium-dependent D2 receptor desensitization and precluded cocaine-induced plasticity’*.*”*

This point has been addressed in the revised manuscript based on the ‘viral expression’ of D2L. It is unclear how the viral expression of D2L-receptors differs from D2L expression in wild type mice. Three points to consider are included in the revised manuscript. First, viral-mediated expression resulted in variable over-expression of D2L receptors, which may preclude or recapitulate cocaine-induced changes. Second, this method may not be subject to the same genetic regulation as endogenously expressed receptors. Third, there may be developmental changes in dopamine neurons that lack D2L receptors that alter the regulation of these receptors when introduced.

“If some DA neurons express either D2S or D2L, they do not show cocaine-induced autoreceptor plasticity, and if D2L blocks plasticity, then it is not clear under what conditions cocaine would induce autoreceptor plasticity (in WT mice).” Another reviewer suggested: “The authors should discuss why this might be the case. For example is it possible that cocaine treatment upregulates D2L in wild type neurons, thus eliminating calcium-dependent desensitization? Could this be tested?”

A previous study suggests that the majority (65%) of dopamine neurons in mouse express both D2S and D2L (10). It is expected that these wild type dopamine neurons are capable of autoreceptor plasticity. However, viral expression of D2L in dopamine neurons that expressed D2S receptors (either viral-mediated or transgenic) did not mimic D2 receptor signaling in dopamine neurons from naïve wild type mice. It is unclear whether this result is due to differential translation or trafficking of endogenous D2L in wild type dopamine neurons (i.e. affecting the ratio of functional D2S and D2L), or is due to developmental changes in dopamine neurons prior to expression of D2L receptors. These caveats are included in the revised manuscript.

It may be that the expression of D2S-only is insufficient for cocaine-induced plasticity because, in wild type dopamine neurons, cocaine treatment upregulates D2L, as described in the Discussion, by an increase in transcription or translation. Both have been described to occur following exposure to D2 agonists (12; 21; 43; 58; 60). Testing these possibilities after in vivo cocaine exposure is currently beyond the expertise of our group. An alternative possibility proposed by many groups (12; 42; 54; 60) is that D2 agonists induce trafficking of D2L receptors preferentially retained in intracellular storage. Testing this possibility is of high interest to our group. We are developing tools that may address this question in future studies. Lastly, it is possible that cocaine treatment upregulates another yet-to-be identified protein that preferentially interacts with D2L or promotes D2L receptor function. These possibilities are now explicitly described in the Results and Discussion.

*“Or alternatively, perhaps the cellular location of D2Rs expressed after viral infection is sufficiently distant from the sites of cocaine action to preclude the drug effect. The spontaneous IPSC data suggest that receptors can be located near sites of DA release, but perhaps the majority of them are not so close. Other factors that could affect this outcome should also be considered.” Although additional tests are suggested, this point could be adequately addressed with further discussion*.

These are good points. Cellular location of D2 receptors may affect cocaine-plasticity. However, the absence of cocaine-induced plasticity in D2S-only neurons was not an artifact of viral-mediated expression. This was examined by testing the effect of cocaine exposure in the transgenic D2-Short mouse line. Dopamine neurons from these mice also did not show cocaine-induced changes (presented in the revised Figure 7). It is possible that the over-expression of D2S receptors interfered with the observation of cocaine-induced plasticity. To examine this possibility, we took advantage of the variable viral expression and only analyzed cells in which quinpirole produced wild type-sized outward currents (∼200 pA with EGTA internal and ∼400 pA with BAPTA internal). The exclusion of large amplitude currents, presumably those over-expressing D2S, did not change the results. The decline in the quinpirole-induced current was greater with EGTA internal compared to BAPTA in both naïve and cocaine-treated mice (naïve: EGTA: 46±4%, n=14; BAPTA: 32±3%, n=7, p=0.01; cocaine: EGTA: 40±3%, n=6; BAPTA: 21±2%, n=5, p=0.01). This point is included in the revised manuscript.

When overexpressed in cultured neurons, D2S and D2L receptors show similar distributions in the somatodendritic compartment (30). But, their density at synaptic and extrasynaptic sites is not known. The sIPSCs data do indicate that both DS and D2L are located at synaptic sites. However, the extent to which these receptors are located at ‘extrasynaptic sites’ is not easily determined. Tonic D2 receptor activation, which may be indicative of extrasynaptic D2 receptor activation, was rarely observed or very small in all of the experiments. In addition, the site of cocaine action necessary for D2 autoreceptor plasticity is not known. It remains to be determined whether this plasticity occurs in the absence of the dopamine transporter (DAT). Even so, the precise localization of D2S and D2L relative to DATs is not known and not easily assayed by electrophysiology since D2 receptors are known to regulate the surface expression of DAT (Bolan et al., 2007). This regulation may be different between the splice variants (Lee et al., 2007). These questions are of high interest, but extend beyond the scope of this manuscript.

A further point of suggested additional discussion concerns the interpretation of the fact that treatments that potentiate D2-activated GIRK current had little effect on D2S expressing neurons. Could D2L receptors be desensitized by endogenous DA to a greater extent than D2S receptors, and could this long-lasting desensitization be reversed by suppressing calcium signaling?

The results demonstrate that depleting intracellular calcium stores or blocking L-type calcium channels increased the magnitude of D2L-activated GIRK currents to a greater extent than D2S-activated GIRK currents. However, it is unlikely that greater tonic D2L desensitization accounts for the difference in GIRK current potentiation. If suppressing calcium signaling reversed D2L desensitization, it is expected that these treatments would have also affected the desensitization to a saturating concentration of agonist, which was not observed. At this point, it is unclear if the small potentiation in D2S-GIRK current is related to removal of desensitization. These results are cautiously interpreted in the revised manuscript where we state that depleting intracellular calcium stores or blocking L-type calcium channels “differentially increased D2S and D2L receptor-dependent GIRK signaling.”

The reviewers also felt that it was important to acknowledge a few caveats. First, presumably cells varied in their degree of transfection and some cells may have expressed receptors at supra physiologic levels. Second, knockout could have produced some compensation.

These caveats are discussed in the revised manuscript.

*Finally, although it was likely most of the neurons patched were DA neurons, it would be important to acknowledge that the neurochemical identity of the neurons was not verified*.

The methods now include an elaboration of the electrophysiological identification of dopamine neurons and also acknowledge that the neurochemical identity of the neurons was not verified.

A couple of other clarifications or amplifications were requested: Were subtype specific effects observed in all cell recorded?

All experiments were carried out blinded to the receptor splice variant. After the initial experiments were un-blinded, it was possible to guess the isoform expressed by subsequent mice after one or two recordings. The results were therefore qualitatively very consistent. Scatterplots of data from the splice variants are also presented in Figures 5 and 6.

How reliable was the cocaine-induced behavioral response and was it accompanied by hyperlocomotion?

No behavioral responses, including hyperlocomotion, were examined.

*It was not clear to one reviewer how L channels were activated in the experiments in*
Figure 6
*and the accompanying supplement. Was the membrane potential held at -60 mV throughout? Do the authors think the relevant portion of the cell escaped voltage clamp?*

In the desensitization assays, neurons were exposed to the L-type blocker, isradipine for >20 min prior to being voltage-clamped at -60 mV. Since dopamine neurons in the SN express low-voltage activated L-type Cav1.3 channels, these channels were likely active during tonic firing. In experiments with iontophoretic application of dopamine, the neurons were voltage-clamped at -60 mV prior to application of isradipine. Once every 50s, prior to the iontophoretic application of dopamine, we made a brief voltage step from -60 to -70 back to -60 mV. A similar step is sufficient to induce calcium entry via L-type calcium channels in dopamine neuron dendrites (Hage and Khaliq, 2014). The revised manuscript includes this methodological detail. However, it is unclear whether this step was necessary to observe the effect of isradipine. While there was no dramatic clamp escape, it is possible that there was a slight deviation in membrane potential. Cav1.3 channels are steeply voltage-dependent between -60 and -50 mV. Thus, at the holding potential used in this study a small activation of these channels is not unexpected.

*Another reviewer was confused about why the authors start using ionotophoretically-ejected DA in*
Figure 5*, but not in the previous figures. Why not more systematically compare saturating and subsaturating D2R activation across the whole paper?*

1) Why not use subsaturating D2 activation to test desensitization? D2 receptor desensitization in response to subsaturating D2 agonist has been described in a previous study (45). The goal of the current study was to induce maximal desensitization as quickly as possible at a concentration of agonist high enough to make irrelevant any possible differences in quinpirole potency at the splice variants. A systematic comparison of saturating and subsaturating D2 receptor desensitization was not conducted across the whole paper since it would have required at least double the number of mice used in this study. This comparison may be warranted for future studies.

2) Why not use saturating D2 activation to test the effect of CPA and isradipine on the magnitude of D2-GIRK currents? Between-group comparisons of the effect of CPA and isradipine on the magnitude of maximal current produced by quinpirole in wild type neurons are now included in the revised manuscript. These results suggest that there is no significant effect of CPA or isradipine on the magnitude of maximal currents produced by bath application of quinpirole. Similar between-group comparisons in D2S and D2L neurons were not made due to variability in amplitude produced by the degree of virus infection. The goals of the iontophoretically applied dopamine experiments were to: a) avoid pre-desensitizing D2 receptors with saturating agonist, b) avoid a ceiling effect that may occur with saturating D2 receptor activation of GIRK channels, c) measure a stable current amplitude before application of drug to allow for within-cell comparisons, and d) enable high temporal resolution of changes occurring over the course of >30 mins. These goals were best met with focal and transient activation of receptors using the iontophoresis of dopamine.

In a previous study, CPA augmented the D2-current produced by repeated submaximal ‘puffs’ of dopamine but not a single submaximal application of quinpirole (45). Thus it is unlikely the difference in our results from wild type neurons is due to differences between saturating and subsaturating application. It may be that dopamine (vs. quinpirole) or repeated agonist application is necessary to observe the augmentation. The mechanism by which the augmentation occurs remains to be determined. These results provide the framework for future studies by demonstrating that D2L-GIRK currents are more readily augmented.

Also, do the authors have evidence that the DA-mediated currents are completely D2R-dependent?

All quinpirole-induced currents and dopamine-mediated sIPSCs were completely abolished by the D2 antagonist sulpiride. In addition, there were no quinpirole (n=16 cells) or dopamine-induced currents in D2 receptor-null dopamine neurons (present study and [5]).

*This reviewer also felt that justification of the high dose of quinpirole was called for*.

D2 receptor desensitization has been assayed with lower concentrations of quinpirole (45). A saturating concentration was used in this study so that the application of agonist would also yield a measurement of the functional expression of D2 receptors. Without such high concentrations, we would have been unable to assess whether the differences observed between D2L and D2S receptors were due to differences in the degree of viral-mediated expression. In addition, as noted above, the use of a saturating concentration obviates issues related to the affinity of D2L and D2S receptors for quinpirole.

*One reviewer suggested: “It would be nice to see stimulus evoked D2/GIRK-mediated IPSCs in addition to the spontaneous synaptic responses shown in*
Figure 1*. This might indicate if the synaptically-accessible D2 pool is similar in the infected neurons*.*”*

A previous study reported that there were no differences in the amplitude or kinetics of stimulus evoked D2-IPSCs mediated by D2S and D2L receptors (Table 2 in [41]). This comment is included in the revised manuscript.

*Finally, one reviewer had major suggestions for improving the presentation which the other reviewers agreed would be beneficial: “The work would be more convincing and easy to follow if the results of experiments with WT DA neurons was compared, in each figure, with the results obtained with neurons expressing D2S only, D2L only or the combination of both. This would help to make clearer how individual isoforms recapitulate the observations made in neurons expressing the native receptors. Such a comparison would be more informative than the comparison performed with the GABAB agonist, which is interesting, but perhaps not as informative*.*”*

Due to the many experimental (e.g. method of expression, developmental changes) and physiological (degree of expression) differences, we are hesitant to make any statistical comparisons with WT, D2S, and D2L. The manuscript was changed in three ways based on this comment. First, all results of current amplitude, desensitization, and augmentation for all groups are plotted in identical formats with the same axes to aid in visual comparisons. Second, wherever possible, data from WT is included with D2S and D2L expressing neurons. Third, the revised Figure 7 now describes the calcium-dependence of D2 receptor desensitization for all groups (control and cocaine-treated WT, D2S, D2L, D2S and L, and transgenic D2-Short).

[Editors' note: further revisions were requested prior to acceptance, as described below.]

*As you can see, one of the reviewers still feels that the conclusion about the role of the D2L in the cocaine-dependent plasticity needs to be toned down and that additional caveats should be discussed. Specifically the reviewer felt that because of the viral strategy used, the possibility of non-cell autonomous effects (e.g. of expressing D2R in GABAergic neurons) could not be ruled out. They also felt that the conclusion about D2L is over stretched, since the experiments do not directly demonstrate the role (restoring D2S is not enough, nor is D2 + D2L). Discussion between the Reviewing Editor and the other two reviewers led to agreement that textual changes in the Discussion to acknowledge these caveats would be sufficient*.

The text of the manuscript has been revised according to the comments and suggestions. In particular, the conclusion about D2L in cocaine-dependent plasticity has been softened. Emphasis is now made on the exclusive expression of D2S as “insufficient” or “not permissive” for cocaine-induced plasticity in D2 receptor-GIRK signaling. Desensitization of D2S was readily reduced by changes in calcium signaling (intracellular stores and L-type calcium channels), but not in vivo cocaine. This was also observed in the transgenic mouse where D2S receptors were selectively expressed using the TH promoter. Thus, the conclusion is made that it is unlikely that cocaine exposure is altering D2 receptor desensitization in wild type dopamine neurons by changing calcium entry via intracellular stores or L-type calcium channels, otherwise it would have been observed in D2S-only neurons. Then the possibility that D2L autoreceptors may have some role in cocaine-induced plasticity in wild type dopamine is suggested, but it is made clear that this has not been directly demonstrated. Additional caveats of non-cell autonomous effects are also discussed.

Reviewer #2:

*The study convincingly demonstrates the role of D2S receptors in calcium-sensitive desensitization and elaborates on two mechanisms*.

However, the argument that restoring D2S expression in D2R KO mice does not confer WT cocaine-induced plasticity, and that it therefore involves D2L receptors is weak, given that viral restoration of D2S and D2L receptors together does not confer the plasticity. There are other explanations for why restoration of D2L does not restore the plasticity, many of which are discussed, but still do not strengthen the conclusion that D2L autoreceptors are required for cocaine-induced plasticity.

*Some of the confounds in the interpretation of the results may have to do with lack of specificity of the D2S and D2L restoration to dopamine neurons (as all cells in the injection site were non-specifically transfected). It would have been cleaner to use D2R KO mice also carrying the DAT*^*IREScre*^
*allele and conditional viruses to restrict D2R subtype expression to dopamine neurons*.

The conclusion that D2L autoreceptors are required for cocaine-induced plasticity has been changed. The revised manuscript now states that the above conclusion has not been directly demonstrated. The data best support the conclusion that the expression of D2S only is insufficient or somehow not permissive to cocaine-induced plasticity. Steps were taken in targeting the viral injection site to the rostral-lateral portion of the SNc to avoid infection of GABAergic neurons in the VTA or SNr. But since neurons were non-specifically transfected, some GABAergic neurons were infected, though the population is likely very small. In the hundreds of neurons patched in this study, only a few infected neurons were found to be GABAergic. This is a potential confound in the interpretation of results from the virus-mediated D2S or D2L studies. However, in support of the conclusion that D2S-only is insufficient for cocaine-induced plasticity, studies were conducted in the transgenic D2-Short mouse (Figures 4 and 7), where expression of D2S was driven by the tyrosine hydroxylase promoter. Results from these mice recapitulated the results where D2S was virally expressed.

The possibility of another type of non-cell autonomous effect is also now addressed in the revised manuscript. Variegated D2 receptor expression inherent to viral infection left some neurons uninfected. The loss of D2 autoinhibition in these neurons may allow elevated release of dopamine, affecting neighboring dopamine neurons. It is possible that, by this mechanism, the combination of D2S and D2L did not resemble wild type dopamine neurons and precluded cocaine-induced plasticity, since neurons may have already been exposed to elevated levels of dopamine.

*Furthermore, the identification of dopamine neurons by physiological criteria is potentially impacted by D2R expression, and may have confounded the results*.

The Methods in the revised manuscript clarifies the point that expression of D2 receptors was not used as a physiological criterion for dopamine neuron identification. The location and electrophysiological properties used to identify dopamine neurons. These well-characterized physiological properties are a reliable method used to identify dopamine neurons in the SNc and were also validated in this study. D2 receptor-dependent currents were produced by quinpirole application in every wild type dopamine neuron identified by these parameters alone.